# Maternal Investment Fosters Male but Not Female Social Interactions with Other Group Members in Immature Wild Spider Monkeys (*Ateles geoffroyi*)

**DOI:** 10.3390/ani13111802

**Published:** 2023-05-29

**Authors:** Carolina Soben, Miquel Llorente, Paula Villariezo, Katja Liebal, Federica Amici

**Affiliations:** 1Fundació UdG: Innovació i Formació, Universitat de Girona, 17003 Girona, Spain; 2Departament de Psicologia, Facultat d’Educació i Psicologia, Universitat de Girona, 17003 Girona, Spain; 3Institute of Biology, Faculty of Life Science, University of Leipzig, 04103 Leipzig, Germany; 4Department of Comparative Cultural Psychology, Max-Planck Institute for Evolutionary Anthropology, 04103 Leipzig, Germany

**Keywords:** social bonds, juveniles, infants, affiliative relationships, maternal care

## Abstract

**Simple Summary:**

In primates, maternal investment is crucial for offspring’s social development, but the benefits it provides may differ depending on their sex. Here, we studied whether female and male immatures in male-philopatric Geoffroy’s spider monkeys (*Ateles geoffroyi*) differ in their social behaviour, and whether they receive different benefits from maternal investment. Our results showed no sex differences in the social development of offspring with regards to body contact and grooming, as both behaviours followed a similar pattern in both sexes during their first six years of life. However, we found sex differences in patterns of social play: the probability of playing with other group members was rather constant throughout age for females, whereas, for males, it became higher than females around two years of age, peaking between three and four years of age. Moreover, there were differences in how mothers fostered social interactions with other group members in female and male offspring: in sons, higher maternal investment was linked to a higher probability of playing with other group members, but this link was not found in daughters. Overall, mothers appear to play a critical role in the social development of immature spider monkeys by fostering the abilities that their offspring will need as adults.

**Abstract:**

In several species, individuals form long-lasting social relationships with other group members, which provide them with important fitness benefits. In primates, patterns of social relationships are known to differ between sexes, but little is known about how these differences emerge through development or the role that mothers might have in this process. Here, we investigated how sex differences in social behaviour emerge during the first six years of primate life and how sex-biased maternal investment can foster immatures’ social development and social interaction with other group members. For this purpose, we observed 20 males and females aged between zero and six years in a wild group of spider monkeys (*Ateles geoffroyi*) that was male-philopatric and, therefore, expected to show sex-biased maternal investment. Our results showed no sex difference in the social development of offspring with regards to body contact and grooming, but the probability of play was rather constant throughout age for females, whereas, for males, it became higher than females around two years of age, peaking between three and four years of age. Moreover, we found differences between female and male immatures in the importance of maternal investment (which included the time mothers spent nursing, carrying, grooming, touching and playing with their offspring) for their social integration in the natal group. In particular, maternal investment increased the probability of playing with other group members for sons, but not for daughters. Our findings suggest that mothers, through sex-biased maternal investment, might have a crucial function in the social development of spider monkeys, fostering the abilities that young offspring need to thrive as adults. By shedding light on maternal investment and social development in a still understudied primate species, these findings contribute to understanding the evolutionary roots of human maternal care and social development.

## 1. Introduction

In several animal species, individuals form preferential affiliative relationships (hereafter, social bonds) with specific group members [1]. Although most animals preferentially form social bonds with genetically related individuals [2,3], several species also form long-lasting bonds with unrelated individuals [1,4]. Across different taxa, the strength and stability of these bonds can have a significant positive impact on individuals’ fitness by increasing their ability to cope with stressful events [5,6,7,8,9], fostering longevity [10,11,12] and reproductive performance [13,14,15,16], and increasing infant survival [17,18]. Social bonds are likely to provide fitness benefits to most group-living mammals, but, so far, most evidence of the adaptive value of social relationships has been provided in primates [1].

In primates, males and females show important differences in how they form and maintain social bonds with other group members. In female-philopatric species, for instance, females usually remain in their natal group after sexual maturity, whereas males tend to leave the group. In these species, females usually form strong social bonds with other group members by frequently engaging in social interactions with specific partners (e.g., white-faced capuchin monkeys, *Cebus capucinus* [19]; rhesus macaques, *Macaca mulatta* [20]; vervet monkeys, *Cercopithecus aethiops* [21]; savannah baboons, *Papio cynocephalus* [22]). In male-philopatric species, in contrast, males usually remain in the natal group and form strong social bonds with their conspecifics, often grooming, playing or maintaining proximity with them (e.g., spider monkeys, *Ateles geoffroyi* [23]; muriquis, *Brachyteles arachnoids* [24]; chimpanzees, *Pan troglodytes* [25,26,27]). In primates, therefore, the philopatric sex usually forms stronger bonds compared to the dispersing sex as they frequently interact with individuals that will likely remain in their same group for many years. 

Sex differences in the strength of social bonds appear to emerge early on during primate development. In several species, for instance, the dispersing sex will have already formed looser social bonds in the first years of life, although sex differences often increase as individuals approach sexual maturity [28,29]. In macaques (*Macaca* spp.), sex differences are already evident in the strength of mother-offspring bonds because mothers form stronger social bonds with philopatric daughters than dispersing males (reviewed in [30]). Immature female macaques are also more likely than males to form social bonds with maternal kin and other females, whereas immature male macaques are more likely to bond with adult males and age peers with whom they will likely migrate into a different group upon reaching sexual maturity [30,31,32]. Crucially, these sex differences in sociality also increase over time as individuals gradually adjust to the species-specific patterns of social interactions and acquire the social roles they will have as adults [31]. Moreover, sex differences in social bonding might also emerge in the social behaviours preferentially used to interact with others. In macaques, for instance, female immatures become gradually more likely than males to groom with conspecifics when approaching sexual maturity, whereas male immatures are more likely to play with other group members throughout development, especially with other males [31]. In female philopatric species, such as macaques, these findings have suggested that play is a functional endeavour for offspring to explore new possibilities of social bonding and is thus especially useful for the non-philopatric sex, with grooming instead being functional for the maintenance of well-established, long-term relationships, and thus especially useful for the philopatric sex [31]. In male philopatric chimpanzees, indeed, sex differences in social development follow a different pattern and are overall less pronounced, with no sex differences in the development of grooming, and play being more frequent in males only during the first couple of years [33]. 

To date, there is still little we know about how sex differences in social bonding emerge. Primates are characterized by prolonged infant dependence, extensive maternal investment and long inter-birth intervals [34,35,36,37], and are thus an ideal model to address this research issue. During the first part of their lives, immatures form the strongest bond with their mothers, who not only provide them with food, warmth, protection and crucial opportunities for social learning [29,38] but may also influence immatures’ social development and their integration into social groups [29,30]. Several studies have indeed shown the importance of maternal investment (i.e., costly behaviours that mothers direct towards their offspring, increasing their fitness [39]) as a crucial component of immatures’ social development. Maternal investment, for example, can have positive effects on the social development of immatures and their integration into the group by fostering the development of the skills and networks they will need to navigate the social world as adults [29,40,41]. Moreover, when mothers interact with other group members in proximity to their offspring, they can also affect offspring’s exposure and access to social partners [29,33,42,43]. Maternal investment can also vary depending on the offspring’s sex, with mothers in good physical condition being expected to invest more in the sex providing higher fitness returns (in polygynous species, sons [44]). In primates, there is some evidence of sex-biased variation in maternal investment (for a review, see [45]), with higher-ranking mothers often investing more in sons than in daughters (e.g., in chimpanzees, *Pan troglodytes* [46]; in black spider monkeys, *Ateles paniscus* [47]). However, the underlying assumption that maternal investment towards sons provides stronger fitness benefits compared to maternal investment towards daughters has rarely been tested in primates [48,49,50]. Moreover, depending on the socio-ecological characteristics of the study groups, there might also be variation in how effectively mothers can increase reproductive success in male and female offspring [48,49]. In primates, for instance, maternal investment may be more effective for the philopatric sex because mothers may better transmit their qualities to individuals that remain longer in the group [48]. In male-philopatric primates, therefore, maternal investment might be higher towards sons than towards daughters, not only because males in polygynous species may generally provide higher fitness returns but also because maternal investment towards the philopatric sex may generally provide stronger benefits as males remain in their natal group.

In this study, we had two main aims. First, we aimed to investigate whether sex differences in social behaviour emerge during the first six years of development in male philopatric spider monkeys (*Ateles geoffroyi*). Although studies in other taxa suggest that sex differences emerge early during ontogeny [30,31], little is known about this species. Second, we aimed to assess whether maternal investment facilitates offspring social interactions with other group members, but only/especially in males, as maternal investment directed towards sons may provide stronger fitness benefits than when it is directed towards daughters in polygynous and male-philopatric species. To address this issue, we studied social behaviour in 20 immatures aged between zero and six years living in a wild group of spider monkeys. In spider monkeys, adult females usually spend less time than males in social interactions with other group members and form weaker relationships with each other; this is in contrast to males, who have stronger bonds with other males and are more cohesive and cooperate regarding defending the territory of the group [35,51,52,53]. We, therefore, hypothesized that immatures’ sex would mediate (i) the social developmental trajectories of their affiliative interactions with other group members, with sex-specific linear and non-linear changes in social behaviour during the first years of life (Hypothesis 1) and (ii) the positive effects of maternal investment on immatures’ probability of interacting with other group members (Hypothesis 2; Table 1). In particular, we predicted that (i) males, being philopatric, would be overall more likely than females to socially interact with other group members, especially through social play [33], with sex differences emerging early on and increasing throughout development as immatures approach sexual maturity (Prediction 1; Table 1; [30,31,32]) and that (ii) higher maternal investment would increase immatures’ probability of socially interacting with other group members, especially in male offspring (Prediction 2; Table 1; [44,48,49,50]). By shedding light on maternal investment and social development in a still understudied primate species, these findings contribute to understanding the evolutionary roots of human maternal care and social development and the patterns that humans, despite cross-cultural variation, share with other primates [54]. 

## 2. Materials and Methods

Ethics. We obtained permission to conduct this study from the CONANP (Comision Nacional de Areas Naturales Protegidas) and SEMARNAT (Secretaría de Medio Ambiente y Recursos Naturales). Our procedures were merely observational and complied with the Principles for the Ethical Treatment of Nonhuman Primates and the Code of Best Practices for Field Primatology by the American Society of Primatologists [55].

Field site and study subjects. We conducted the study on a group of 49 well-habituated Geoffroy’s spider monkeys (*Ateles geoffroyi*) in the natural protected area called Otoch Ma’ax Yetel Kooh in Yucatan, Mexico (20°38′ N, 87°38′ W). The group was composed of 13 adult females, 9 adult males, 3 subadult females, 2 subadult males, 9 juvenile females, 5 juvenile males, 3 infant females and 5 infant males (see [56]; Table 2). We could individually recognize the monkeys using differences in their facial characteristics, size, genital features and fur colouration. Individual age and mother-infant relationships were established thanks to the long-term demographic records of the study group.

Data collection. We conducted behavioural observations from August 2021 to June 2022. We collected data 5 days a week, from 06:00 to 13:30, using 15-min focal animal samples and continuous sampling [57]. To record the observations, we used CyberTracker, with one to two observers dictating the behaviours observed, and a third observer entering the information in CyberTracker. This approach allowed one observer to monitor the group and avoid losing it, and one observer to continuously monitor the focal subject in the canopy, without losing sight of the focal when the third observer was entering data. We observed all the individuals who were younger than 6 years old (Table 2), pseudo-randomizing the order in which they were observed (i.e., observing the first available individual from a list in which all immatures were randomly ordered), without sampling subjects more than once per hour. During the focal samples, we recorded (i) the duration of the focal sample (i.e., excluding the time in which the focal individual was out of view) and whenever the following interactions occurred with individuals other than mothers, (ii) the duration of each body contact event involving the focal animals, (iii) giving or receiving grooming and (iv) playing (i.e., exaggerated spontaneous behaviour between two or more individuals including object play, acrobatic play, sparring and wrestling [58]), and partner identity.

From February to June 2022, during focal samples, we further recorded the duration of time the immature’s mother spent (v) carrying, (vi) nursing, (vii) grooming, (viii) touching or (ix) playing with the immature as proxies of maternal investment. In contrast to the behaviours listed above (ii to iv, Table 3), which relate to interactions between immatures and group members other than mothers, these behaviours (v to ix, Table 3) exclusively refer to the interaction between mothers and their immatures. The choice to monitor these behaviours was based on the literature on maternal investment and mother–infant relationships in primates (e.g., [45,59,60,61]). Although data on maternal investment were collected for a more limited amount of time than data on immatures’ social behaviour, we consider 5 months to be a long enough timespan to provide a representative measure of maternal investment. We further recorded (x) the duration of time mothers spent within 1 m of the immature, as proximity to mothers might expose immatures to more group members and facilitate interactions with them [29,33,42,43]. Given that the mother of 2 immatures disappeared from the group and we thus could not collect any information on maternal investment, those 2 immatures were removed from the analyses. This resulted in a sample of *N* = 20 study subjects (Table 2) and a total of 623 focal samples (mean ± SD: 7.8 ± 1.83 h per focal animal). Finally, to assess social relationships with other group members, we also conducted 15 min focal animal samples on all the other individuals in the group (Table 2), measuring (xi) the time the focal animal in view was giving or receiving grooming, and the partner identity. Regardless of their age, during each focal sample, we also recorded (xii) the identity of all group members within a 5 m proximity from the focal, with a 2 min interval. 

Statistical analyses. First, we operationalized maternal investment towards immatures by calculating for each infant and juvenile the mean proportion of focal time that mothers spent carrying, nursing, grooming, touching and playing with the immature. As we wanted to obtain a single maternal score including all these 5 measures, but some of these behaviours were much more frequent than others, we rescaled them so that each of the 5 measures could vary between 0 and 1, thus contributing in a comparable way to the measure we used as a proxy of maternal investment (i.e., the average of these 5 rescaled measures). 

We then used R [62] and the package glmmTMB [63] to run generalized linear mixed models. In the dataset, we entered one line for each focal observation (*N* = 623). Our dependent variables were whether the focal animal spent time in bodily contact (Model 1), grooming interactions (Model 2) or playing (Model 3) with an individual other than the mother. All models were run with a binomial distribution. As test predictors in all models, we included the 2 2-way interactions of immature’s sex with immature’s age and immature’s sex with immature’s squared age (as the link between age and response may not have been linear; e.g., [31]) to test Prediction 1 (Table 1) and the 2-way interaction of immature’s sex with maternal investment to test Prediction 2 (Table 1). As a test predictor, we also included whether the mother was within 1 m proximity of the immature during the focal observation as this might facilitate interactions with other group members. In the models, we also included the terms of the 2-way interactions as main effects and the duration of the focal observation as an offset term; we also controlled for seasonality (i.e., whether the focal observation was conducted during the dry or wet season [64]) and we included immature identity nested in mother identity as a random factor (i.e., we included both mother and offspring identity as random factors, but nested, as each offspring only had one mother, whereas some mothers had more than one offspring). We z-transformed continuous predictors (i.e., immature’s age and maternal investment) to facilitate model convergence and the interpretation of model coefficients. We compared the full models described above to corresponding null models that were identical but did not include test predictors [65]. If the full models differed from the null one, we used the drop1 function to assess the significance of the test predictors, assigning significance to values that were less or equal to 0.05. We checked model assumptions with the “DHARMa” [66] and “performance” packages [67]. We checked overdispersion and multicollinearity with the functions check_collinearity and testDispersion, and they were not an issue (maximum variance inflation factors across models = 3.29 [68]). We detected no zero-dispersion and no convergence problems for any of the models presented. Models using proportional responses (i.e., the proportion of focal time immatures spent in body contact, grooming interactions or playing), rather than binomial ones, however, failed to converge, and are not reported here. We did not check for autocorrelation in our dataset, as we only conducted up to one focal observation for each individual and day. 

Finally, we ran social network analyses [69]. First, we built an undirected weighted matrix for grooming interactions and one for proximity as both these behaviours were observed in all group members. For each possible dyad in the group, we entered the proportion of time in which the 2 individuals engaged in the social behaviour out of the total time the 2 individuals had been visible as focal animals, assigning 0 to all mother–infant dyads. We then ran social network analyses on these matrices, using the following packages in R: vegan version 2.5–3 [70], asnipe version 1.1.10 [69], and igraph version 1.2.1 [71]. We then extracted individual values of eigenvector centrality for grooming and proximity, which were scores between 0 and 1 (0 being assigned to the least socially integrated individuals) that measured the importance of individuals as “social hubs” [72,73]. We finally used exact Spearman correlation tests to assess whether eigenvector centralities of male and female immatures correlated with those of their mothers.

## 3. Results

For Model 1, the full model did not significantly differ from the null model (GLMM, χ^2^ = 14.40, df = 8, *p* = 0.072; Table 4), suggesting that none of the test predictors included could reliably predict immatures’ probability of being in bodily contact with group members other than the mother. For Model 2, in contrast, the full model significantly differed from the null model (GLMM, χ^2^ = 22.26, df = 8, *p* = 0.004; Table 4), with the probability of grooming with other group members increasing for both sexes with mother’s proximity (*p* = 0.006) and increasing immatures’ age (*p* = 0.026; Figure 1).

Finally, for Model 3, we found a significant difference between the full and null models (GLMM, χ^2^ = 20.09, df = 8, *p* = 0.010; Table 4), with a significant effect of the two-way interactions of immature’s sex with squared immature’s age (*p* = 0.048) and with maternal investment (*p* = 0.017). In particular, the probability of playing with other group members remained rather constant for females during the first six years of their life, but it quickly increased for males, peaking around three years of age before decreasing again (Figure 2). Moreover, the probability of playing with other group members increased with maternal investment for male immatures, but not for female immatures (Figure 3).

Social network analyses revealed that female and male immatures were on average in the proximity of 22 and 24 group members other than their mothers, respectively, and grooming with 3 and 1 (Figure 4). After sexual maturity, females and males were on average in the proximity of 21 and 26 group members, respectively, and grooming with 2 and 5. Eigenvector centralities of immatures correlated with that of their mothers for females (Spearman tests, proximity: *rho* = 0.732, *N* = 11, *p* = 0.014; grooming: *rho* = 0.649, *N* = 11, *p* = 0.036; Figure 5) but not for males (Spearman tests, proximity: *rho* = 0.092, *N* = 9, *p* = 0.818; grooming: *rho* = −0.468, *N* = 9, *p* = 0.207).

## 4. Discussion

Partially in line with our hypotheses, we found sex differences in the social development of immature spider monkeys and in the effect of maternal investment on the probability of interacting with others, but only for play (Table 1). In particular, the probability of playing remained rather constant for females during the first six years of their life, but quickly increased for males, with a peak around three years of age. Moreover, maternal investment was linked to a higher probability of playing with others, but only in male offspring.

Our study revealed sex differences in social development, but only for play behaviour. During the first six years of their lives, in particular, immatures’ probability of being in bodily contact with group members other than their mothers did not change, whereas the probability of grooming with them gradually increased in a similar way for both sexes. For play, the probability of playing with other group members remained rather constant during development for females, but not for males. In males, the probability of playing quickly increased during the first years of their lives, becoming higher than in females from the age of two, in line with our first hypothesis (Table 1), and peaking around three years of age. When approaching sexual maturity, however, sex differences in the probability of play disappeared, in contrast with our first hypothesis (Table 1). At least for males, these results are in line with the literature suggesting a peak in primate social play around two years of age (e.g., in chimpanzees: [33]; in macaques: [31]. Moreover, our results are in line with findings in another male philopatric species, chimpanzees, in which grooming also increases in a similar way throughout development for both sexes, and sex differences in developmental patterns of social play are limited in time, decreasing when immatures approach sexual maturity [33]. Although these preliminary findings will need to be confirmed by the inclusion of more species, they suggest variation in how sex differences in social development emerge in male- and female-philopatric species. In particular, the gradual increase in grooming rates that characterizes female development in female-philopatric species appears to be absent in species characterized by male philopatry, where males form longer-lasting relationships and are more likely to interact with each other during adulthood (in spider monkeys: [35,51,52,53]). Similarly, sex differences in social play appear to be more limited in male- than in female-philopatric species: in female-philopatric species, migrating males are more likely than females to play throughout development [31], but in male-philopatric species, these differences seem to decrease when approaching sexual maturity, possibly because males that remain in their natal group do not need to rely as much as migrating males on play to explore new possibilities of social bonding [31]. Like most group-living mammals, the majority of primates show female philopatry [74]. Although the evolutionary reasons for sex-biased dispersal are yet unclear [74,75], it is likely that female philopatry is the ancestral trait, whereas male philopatry evolved as a derived trait afterwards [76]. Therefore, it is possible that the social patterns that evolved for female philopatry (higher grooming rates in philopatric females, higher play rates in migrating males) have been adjusted in male-philopatric species, but have not (yet) fully reversed: grooming increases in a similar way throughout development for both sexes and sex differences in the emergence of social play decrease when immatures approach sexual maturity.

Our study also revealed sex differences in the ability of maternal investment to predict offspring’s social interactions with other group members, in line with our second hypothesis (Table 1). In particular, maternal investment increased the probability that immatures played with others, but this effect was limited to male offspring. In contrast, we found no effect of maternal investment on immatures’ probability of being in bodily contact and grooming with conspecifics. In polygynous species, males are expected to provide higher fitness returns than females [44]. However, very few studies have experimentally assessed whether maternal investment really provides stronger fitness returns when directed towards sons rather than towards daughters [48,49,50]. Our results provide support for this assumption by suggesting a sex bias in the effectiveness of maternal investment, which more effectively fosters social relationships of male rather than female immatures with other group members. Given that social integration provides crucial fitness benefits to primates [5,6,7,8,9,11,16,17,18], our findings suggest that maternal investment towards sons results in a higher increase in fitness for spider monkeys compared to maternal investment towards daughters. In male-philopatric primates, such as spider monkeys [51], maternal investment might also provide stronger benefits to male immatures because of their dispersing patterns. If males remain longer in the group, maternal investment may be more effective towards sons, whereas the contrary might happen in female-philopatric species. Therefore, it is possible that in female-philopatric primates, sex biases in the effectiveness of maternal investment are more difficult to detect [45] simply because maternal investment provides higher fitness returns when directed towards sons [44], but also when directed towards the philopatric sex. Future studies in other male- and female-philopatric species will thus be essential to test this hypothesis.

Unfortunately, our study does not allow us to infer the mechanisms through which maternal investment may facilitate offspring’s social integration. One hypothesis is that mothers who often engage in social interactions with their offspring (which we considered an aspect of maternal investment [45,59,60,61]) might also be in general more sociable and thus more likely to interact with other conspecifics, providing immatures with more exposure to potential partners and thus more opportunities to socially interact with others. Therefore, maternal investment would foster immatures’ social interactions with other group members simply because more sociable mothers, by investing more in their offspring per definition, are also likely to facilitate contact with potential partners. In our study, indeed, proximity to mothers increased the probability of immatures grooming with other group members, but that was not the case for bodily contact and play. Moreover, this explanation is unlikely for two further reasons. First, in our study, we operationalized maternal investment as a multi-component index, including measures of social interactions between mothers and immatures (i.e., grooming, playing) but also other behaviours (i.e., carrying, nursing, touching). Therefore, higher maternal investment does not necessarily mean that mothers are more sociable. Second, social network and correlation analyses showed that maternal social integration correlated to that of female immatures, but not of male immatures (both in terms of proximity to and grooming with other group members). Therefore, it is unlikely that maternal investment fosters males’ social interactions only by increasing their exposure to other group members. Overall, our findings rather suggest that mothers who invest more in their male offspring (by carrying, nursing, touching and socially interacting with them) also foster their social integration, independently of their own social integration, thus providing them with crucial fitness benefits. In contrast, social integration for daughters appears to be more tightly linked to the social network size of their mothers, with no clear role of maternal investment.

If maternal investment in sons really provides higher fitness returns than investment in females, as our findings suggest, mothers in spider monkeys should also invest more in males [44]. Our study does not address this question, but preliminary analyses and previous studies suggest that this might be the case. For example, studies carried out with *Ateles paniscus* have also shown that mothers invest more in sons than in daughters when they are fit. Symington [47], for instance, found that higher-ranking mothers, who are usually in better physical condition than lower-ranking ones, invested more in male offspring than lower-ranking mothers because birth intervals were longer after the birth of male offspring compared to female offspring. Previous work on the same group of spider monkeys observed in our study also suggests that mothers are more likely to invest in male offspring than female offspring, especially when mothers are more experienced and better integrated into the social group [77].

Our study has several limitations. The first limitation is the small sample size, including the fact that individuals were only followed for a few months. More data and longitudinal studies are essential to confirm the results. This limited sample size might explain why sex differences in social development and the link between maternal investment and immatures’ social integration were only found for play but not for other behaviours. In spider monkeys, social play is often used by males to interact with age peers and other partners [31], but grooming is not frequently used [78,79]; thus, more data might be needed to detect sex differences in this behaviour. Moreover, our limited dataset did not allow for exploring how immatures’ social networks gradually grow throughout development, although it is likely that, like in other species [31], patterns of social interactions are also modulated by the focal and partner’s characteristics (e.g., sex, age, rank, etc.). In the future, it will thus be crucial to assess how the characteristics of immatures´ social partners change when immatures grow, and the facilitating role that kin (e.g., brothers, sisters [30]) might have in this process. Furthermore, longitudinal studies following individual developmental trajectories over several years would surely provide a better methodological approach by also allowing the use of other statistical tools, such as structural equation modelling, to simultaneously estimate multiple relationships and allow inferences about possible causal pathways [80]. Importantly, larger sample sizes would not only allow for the detection of weaker effects on the study variables but would also allow the inclusion of more test and control variables in the analyses (e.g., maternal rank and social condition, previous experience as mothers, characteristics of social partners) and for the comparison of groups exposed to different socio-ecological conditions. Furthermore, in our study, we largely measured maternal investment and immatures’ social behaviour over the same period, but it is possible that maternal investment provides fitness benefits only in the longer term. In this regard, it would also be essential to obtain more precise measures of fitness and maternal investment, including the physiological assessment of maternal costs (e.g., measures of C-peptide levels in the urine, food availability or maternal weight of mothers) for the different components of maternal investment [39,81,82]. Finally, future studies should ideally assess whether sex differences in maternal investment depend on a different disposition of mothers towards sons and daughters or are rather triggered by sex differences in offspring behaviour. Assessing who initiates bodily contact, for instance, or whether nursing is solicited, will be highly informative to address this question. Despite these important limitations, our work, as a pilot study, provides an initial contribution to the study of sex differences in the social development of wild spider monkeys and the role of maternal investment in fostering the social integration of young offspring.

## 5. Conclusions

Overall, our findings are suggestive of sex differences in the social development of offspring with regard to social play and confirm that mothers may have a crucial function in the social development of spider monkeys by fostering social integration through sex-biased maternal investment. These results provide novel information about primate social development and maternal care in a male-philopatric species and thus contribute to creating a more comprehensive framework to understand the evolutionary origins of human maternal care and social development.

## Figures and Tables

**Figure 1 animals-13-01802-f001:**
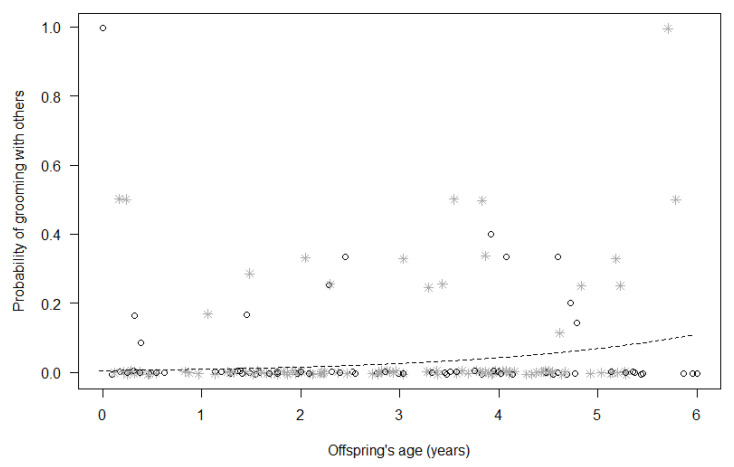
Probability that immatures groomed with group members other than their mothers, as a function of their age (*p* = 0.026). For each study subject, black circles represent the mean monthly probability of grooming for male offspring, whereas grey asterisks represent the mean monthly probability of grooming for female offspring. Data points are slightly jittered to avoid overlapping. The line represents the fitted model, which is like Model 2, with observational effort expressed in 15 min intervals.

**Figure 2 animals-13-01802-f002:**
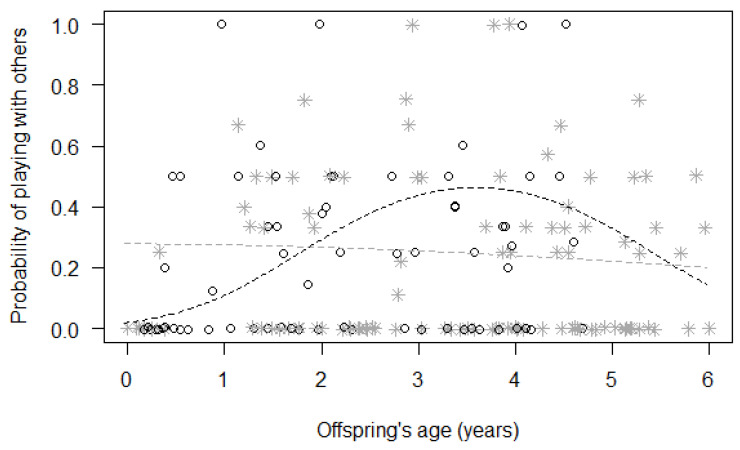
Probability that immatures played with group members other than their mothers, as a function of their squared age (*p* = 0.048) (separately for male and female immatures). For each study subject, black circles represent the mean monthly probability of grooming for male offspring, whereas grey asterisks represent the mean monthly probability of grooming for female offspring. Data points are slightly jittered to avoid overlapping. The two lines represent the fitted model (the black one for males, the grey one for females), which is like Model 3, with observational effort expressed in 15 min intervals.

**Figure 3 animals-13-01802-f003:**
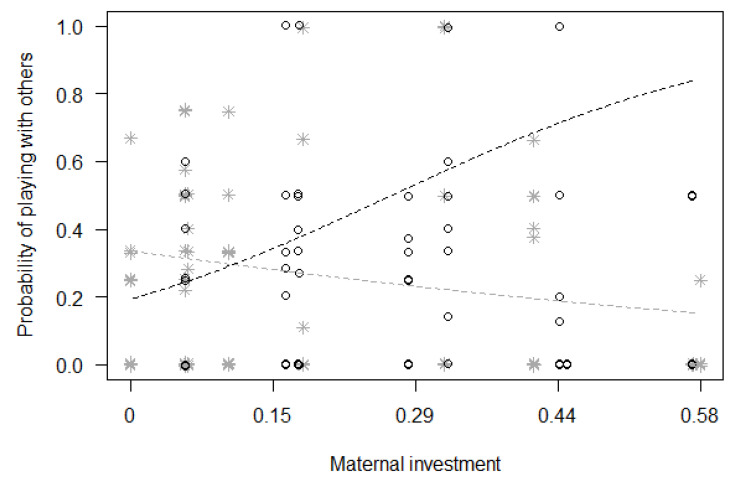
Probability that immatures played with group members other than their mothers, as a function of maternal investment (i.e., a score between 0 and 1, based on the proportion of time mothers spent nursing, carrying, grooming, touching and playing with their offspring, with 0 meaning no maternal investment; see text for more details; *p* = 0.017) (separately for male and female immatures). For each study subject, black circles represent the mean monthly probability of grooming for male offspring, whereas grey asterisks represent the mean monthly probability of grooming for female offspring. Data points are slightly jittered to avoid overlapping. The two lines represent the fitted model (the black one for males, the grey one for females), which is like Model 3, with observational effort expressed in 15 min intervals.

**Figure 4 animals-13-01802-f004:**
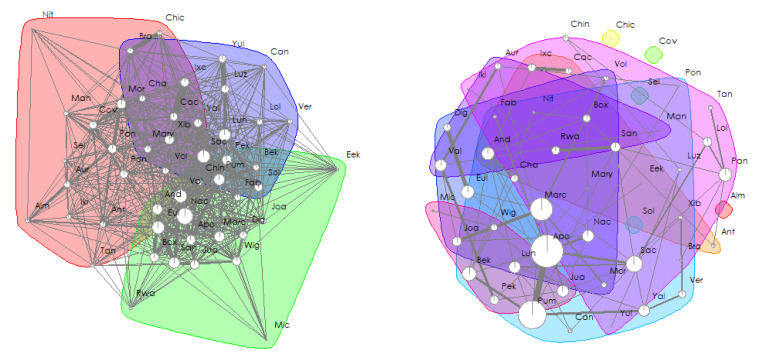
Social networks of the spider monkey group observed, based on the undirected, weighted matrix of proximity (**left**) and grooming interactions (**right**) after removing interactions between immatures and their mothers. Dots represent individuals of the group and are the nodes of the social network, whereas lines represent their edges. The thickness of weighted edges and the size of the nodes are proportional to the individual’s strength in the social network (i.e., the sum of all edge weights connected to the node). Communities (i.e., groups of nodes including a high proportion of the edge weight, detected using leading eigenvector communities) are depicted in different colours (Farine & Whitehead, 2015 [73]).

**Figure 5 animals-13-01802-f005:**
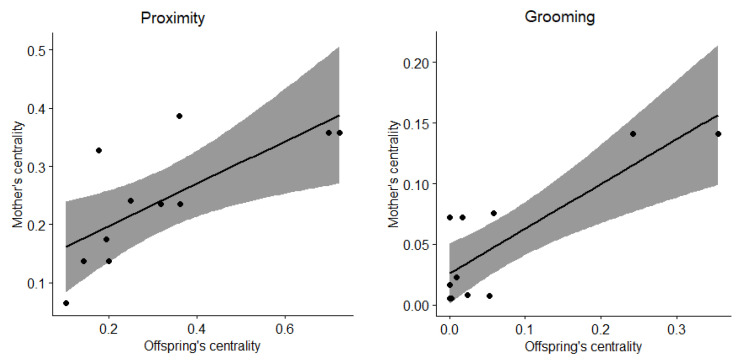
Correlations between eigenvector centralities of female immatures and their mothers for proximity (**left**) and grooming interactions (**right**). Dots represent female immatures, the line represents the regression line and the grey area represents the confidence intervals.

**Table 1 animals-13-01802-t001:** Predictions of our study for the responses measured and whether they were confirmed by the models we conducted. The interactions between predictors are marked with an asterisk (*).

Predictions		Response	Model	Confirmed?	Significant Test Predictors
1	In immatures, males are more likely than females to interact with others, and sex differences increase with age	Body contact	1	No	-
		Grooming	2	No	Immature’s age
		Playing	3	Partly	Immature’s sex * Immature’s age^2^
2	Maternal investment increases the probability of interacting with others, but more so in male immatures	Body contact	1	No	-
		Grooming	2	No	-
		Playing	3	Yes	Immature’s sex * Maternal investment

**Table 2 animals-13-01802-t002:** List of individuals present in the study group, including age class (i.e., infants, <2 years; juveniles, 2–5 years; subadults, 6–7 years and adults, >8 years); sex; age at the onset of the study (for infants and juveniles, in months, for subadults and adults, in years); and mother identity (only for infants and juveniles). Our study subjects were all the infants and juveniles listed below, with the exception of the two immatures marked with an asterisk (i.e., Puma and Yalit), for which information about maternal investment was not available.

Age Class	Subject	Sex	Age	Mother
Infants	Alma	Male	1 month	Antena
	Cacao	Male	4 months	Xibalba
	Chaac	Male	16 months	Ikil
	Chikich	Female	14 months	Tanga
	Covid	Male	5 months	Pancha
	Selva	Female	2 months	Marylin
	Sol	Male	3 months	Joanne
	Yuli	Female	16 months	Lola
Juveniles	Aura	Female	40 months	Antena
	Braga	Female	45 months	Tanga
	Canela	Female	27 months	Veronica
	Eek	Female	55 months	Mich
	Fabrizio	Male	18 months	Rwanda
	Ixchel	Female	63 months	Xibalba
	Luna	Female	28 months	Joanne
	Luz	Female	47 months	Lola
	Pekin	Male	41 months	China
	Poncho	Male	47 months	Pancha
	Puma *	Male	21 months	Flor
	Sacbe	Female	56 months	Joanne
	Voldemort	Male	34 months	Mandíbula
	Yalit *	Female	49 months	Flor
Subadults	Bekech	Female	7 years	
	Morita	Female	6 years	
	Nacho	Male	7 years	
	Nit	Female	7 years	
	Valentín	Male	6 years	
Adults	Andrés	Male	8 years	
	Antena	Female	15 years	
	Apolo	Male	8 years	
	Boxhuevos	Male	11 years	
	China	Female	39 years	
	Digit	Male	9 years	
	Eulogio	Male	18 years	
	Ikil	Female	8 years	
	Joanne	Female	24 years	
	Juan	Male	19 years	
	Lola	Female	21 years	
	Mandíbula	Female	18 years	
	Marcos	Male	15 years	
	Marylin	Female	13 years	
	Mich	Female	13 years	
	Pancha	Female	20 years	
	Rwanda	Female	9 years	
	Sancho	Male	10 years	
	Tanga	Female	20 years	
	Verónica	Female	38 years	
	Wiguiberto	Male	12 years	
	Xibalba	Female	9 years	

**Table 3 animals-13-01802-t003:** List of behaviours, definitions and the period in which behaviours were collected.

Behaviour	Definition	Period
Immatures with group members other than their mother		
Grooming	Manipulation of another group member’s fur by the immature or manipulation of the immature’s fur by another group member.	August 2021 to June 2022
Playing	Exaggerated spontaneous behaviour by the immature and at least another individual, including object play, acrobatic play, sparring and wrestling.	
Cofeeding	Feeding involving the immature and another group member within a 1 m proximity in the same tree.	
Immatures with their mothers		
Carrying	Maternal displacement, while the immature clings to the mother’s back.	February to June 2022
Nursing	The immature holding the mother’s nipple in their mouth.	
Playing	Exaggerated spontaneous behaviour, including object play, acrobatic play, sparring and wrestling.	
Touching	The mother placing the palm of one or both hands on the offspring’s body.	
Grooming	Manipulation of the immature’s fur by the mother.	
Proximity	The immature and the mother being within a 1 m proximity of each other.	

**Table 4 animals-13-01802-t004:** Results of the three best models, with estimates for each predictor, standard errors (SE), confidence intervals (CIs) and *p* values for test predictors (reference category in parenthesis and significant test predictors marked with an asterisk, and two asterisks for strong significance).

Models and Predictors	Estimate	SE	2.5% to 97.5% CIs	*p*-Value
Model 1: The probability of immatures being in bodily contact with group members other than mothers				
Intercept	−8.99	0.24	−9.47 to −8.52	-
Immature’s sex (male)	0.17	0.25	−0.32 to 0.66	0.504
Immature’s age	0.11	0.21	−0.30 to 0.52	0.610
Maternal investment	0.08	0.20	−0.32 to 0.47	0.699
Maternal proximity	0.62	0.28	0.07 to 1.17	0.027 *
Seasonality	−0.09	0.24	−0.57 to 0.39	0.720
Model 2: The probability of immatures grooming with group members other than mothers				
Intercept	−10.47	0.46	−11.37 to −9.58	-
Immature’s sex (male)	0.29	0.50	−0.69 to 1.27	0.566
Immature’s age	0.83	0.37	0.10 to 0.29	0.026 *
Maternal investment	−0.17	0.41	−0.97 to 0.63	0.672
Maternal proximity	1.10	0.40	0.31 to 1.89	0.006 **
Seasonality	−0.51	0.41	−1.32 to 0.29	0.199
Model 3: The probability of immatures playing with group members other than mothers				
Intercept	−7.86	0.24	−8.34 to −7.39	-
Immature’s sex (male)	0.76	0.33	0.12 to 1.40	-
Immature’s age	−0.11	0.22	−0.55 to 0.32	-
Immature’s squared age	−0.03	0.16	−0.35 to 0.29	-
Maternal investment	−0.31	0.22	−0.75 to 0.12	-
Immature’s sex (male) * Immature’s age	0.79	0.44	−0.06 to 1. 64	0.067
Immature’s sex (male) * Immature’s squared age	−0.73	0.38	−1.47 to 0.01	0.048 *
Immature’s sex (male) * Maternal investment	1.24	0.54	0.19 to 2.30	0.017 *
Maternal proximity	−0.33	0.23	−0.79 to 0.13	0.152
Seasonality	−0.37	0.21	−0.78 to 0.03	0.070

## Data Availability

Data are provided as Appendix A.

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
