# Peer review of "Maternal Investment Fosters Male but Not Female Social Interactions with Other Group Members in Immature Wild Spider Monkeys (Ateles geoffroyi)"

_animals, 2023, doi:10.3390/ani13111802_

Round 1

Reviewer 1 Report

The authors investigated sex-differences in the social development involving other group members than the mother during early life in wild spider monkeys, a male-philopatric primate species, and whether the level of maternal investment is linked to levels of social integration, predicting a biased investment towards male offspring. They find that higher maternal investment in sons is associated with more time spent playing and grooming with others in sons, a relationship that is absent for daughters. Overall, the authors found no sex difference in the time spent interacting with other group members of young spider monkeys. While I love the study idea and its potential contribution, I have some major concerns and hope they can be addressed:

The data chosen strongly limits what can be learned from this study about this male-philopatric species and are sufficient to answer the questions appropriately. For example, 1) social integration measures should involve the number (and ID) of different social partners (ideally a social network analysis using association patterns of offspring), and 2) the social environment of the mother as an indicator of offspring exposure to other group members was not assessed, but likely matters, especially while offspring still depend on mothers during travel. It is also unclear how the studied maternal behaviors are expected to translate into social integration/social interaction of offspring with other group members. If the authors have such data, I would strongly recommend to include them in this study. If such data are not available, the authors need to interpret findings much more carefully and acknowledge limitations that are not yet addressed in the discussion. Also, the theory of how the studied maternal investment measures may translate to more time socializing with others needs to be strengthened and backed up with literature. In the case that no additional analysis can be run, I would recommend to avoid talking about social integration per se and prefer social interactions with others.  

I also have some other suggestions/questions especially related to data analysis, which are listed in more detail below:

Detailed comments/suggestions/questions:

Abstract

L29: maybe specify and say: “…for their social integration in the natal group”, because daughters also must succeed in integrating into the social group they disperse to.

L30-33: I would say a bit more in the abstract about what maternal investment behaviors you have used/measured as an indicator of facilitating increased socializing with other group members?

L35: write: young male offspring need

L36: I would argue that daughters also need such skills in adulthood. Therefore, I feel you need to be more specific here and say that you address the social integration (interactions) in the natal group (e.g. to successfully establish their social role and integration into the social network of their natal group).

L112-14: This sentence is a bit redundant (male-philopatric – we know this already) and since you focus on primates mostly in your introduction, maybe use this argument for why primates (113-114) earlier where you narrow down your introduction from social animals in general to primates (e.g. move to L49).

Introduction

L44: maybe say high strength and stability of these bonds

L45: maybe “by increasing the ability…”?

L51&56: minor: female-philopatric /male-philopatric (to keep style consistent across the manuscript following the writing style in the simple summary)

L62&64&65: Same here - minor: either sex differences or sex-differences! Keep style consistent!

L74: “How these differences emerge, however, is not clear.” à maybe adjust and write is understudied.

L83-86: Do you address the same behavioral mechanism facilitates developing social bonds through the mother as described in the previous sentences (mother’s proximity to other group member)?

L92-94: maybe say why this has been rarely tested in primates.

L102: maybe list some of these stronger benefits.

L107-8: Could you be more specific what exactly you want to study in aim 2? Just the link or whether maternal investment is driving social integration?

L118: I would provide information about the age-range of immatures you will cover (5 years, right?) also for the second part of the hypothesis (first part you focus on the first year in life). In fact, the age-range addressed here are not really playing a role in your analysis.

L119-122: your prediction does not match the age you outlined as focus in your first aim and your first hypothesis (first year only à but prediction addresses first year and beyond). Please clarify!

Table 1: Because this is a summary of your findings, it should be moved to the result section. Also, some columns will make more sense once the reader has gone through data analysis and model variables in the method section.

General comments to introduction: The introduction could be strengthened by adding relevant information, e.g. species-specific information about the importance of female-female bonds before/after dispersing (e.g. are there any benefits for fostering female-female bonds if they likely transfer together into a neighboring group at sexual maturity?). Maybe also mention in background information when spider monkey infants become more independent in movement from mothers (being in contact most of the time vs. leaving mom and being able to chose their social partners independently). Finally, can you provide information about what is known about interactions between siblings or relatives (could be also discussion as another potential factor that affects social interactions with non-mother group members). To strengthen your argument about sex-differences, anything known about sex-biased maternal investment in the study species may further help to set up your objectives.

Methods

L139: maybe “using differences” instead of “thanks to”?

L144: the way the age-ranges are written it excludes 5-6 years old monkeys

Table 2: Personally, I find days for infants and juveniles difficult to translate in years and would prefer providing either months or years (1 decimal places).

L151-2: Maybe it is clearer if you write: 15min continues sampling of behavior during focal follows.

L152-4: How does data collection of 2 observers at the same time following one focal exactly work? Can you explain more and why did you choose such an unusual way to collect focal data?

L158-59: suggestion (i) focal animal in view (….)

(ii) to (iv): So this is all about interactions between offspring and other non-mother group members, right? Is a bit difficult to understand but becomes clear when reading on. Maybe rephrase the sentence a bit so this is clearer? Did you also collect the ID of the interaction partner? This latter information is crucial for measuring social integration. In addition, the number of older siblings in the group could also affect interactions with other group members and recording the ID of social partners allows to control for it.

L162: add comma after samples.

L164: Is it possible or appropriate to collect duration of touching? It is not exclusively different from body contact either and seems more of a short behavior that should be presented as frequency?

L157-64: again, some of your behaviors are not exclusive, e.g. grooming requires body contact and so does often playing…I think more detailed definitions of your categories would help.

General comments to method: Your second aim is “to assess whether maternal investment is linked to an increase in offspring’s sociality to other group members”. If you mean here how mothers facilitate sociality to other group members, then I do not understand well how the data you have collected can aid achieving this goal. Is your assumption that mother’s level of investment in her offspring translates directly into social strength and relationships with other group members? While you do gather data on both 1) interaction between mother and offspring and 2) between offspring and other group members, I miss key data as indicator in how far the mother was facilitating the social development with other group members, e.g. as you mention in your introduction: did the mother’s social environment (seeking proximity with other group members/playmates) help stimulate social development and get access to social partners? That would e.g. require data on associations between mother and other group members and how association patterns change across infancy/juvenilehood (e.g. while offspring is still depend on transport by mothers during movement and when is becomes more independent during moving could influence the role of the mother) and whether there are differences in case of having a son vs a daughter. Also, social integration should also consider with how many different other group members the offspring interacts; e.g. an offspring may be involved in play and groom with others very often but always with the same group member (maybe even the same sibling). Since you focus on a male-philopatric species, even the sex of group members with which male and female offspring interact could be very relevant and help understand your findings. I feel that you miss key data which allow you to address your second goal more comprehensively and properly understand what is going on.

L170-174: Did you calculated the mean DAILY(?) proportion of these interactions? I am also not sure whether I fully understand what and why you needed to rescale? Aren’t proportions numbers from 0-1 anyway? And then, you calculated a mean again – of what? How you transformed these 6 measures is a bit confusing. Can you try to clarify? Also, doesn’t time spent with the mother correlated negatively with time spent with other group members? Thus, in your models, how can high maternal investment predict higher sociality of offspring with others? Is this possible?

L174-176: this sentences should be moved up to where you introduced the behavior in your methods.

L178: Did you present time as proportion or in min/sec?

L178: write: Our independent variables were mean proportion of time spent in body contact, grooming, co-feeding, and playing…

L183: do you know of a references that supports your assumption that the relationship may not be linear? If yes, please back it up!

L186: did you use the log focal duration as offset term?

L193-94: explain how you analyzed multicollinearity and dispersion in R (functions/reference)

L195-98: That is very confusing! I think this will be clearer once you explained the actual unit of your dependent variable.

General comments to statistical analysis: 1) Did you also consider GAMMs which can be advantageous when analyzing timespans (age)? 2) Did you check data for temporal autocorrelation and zeroinflation? I can imagine you have quite some zeros in your dataset and focal data collected closely to each other in time may also provide more similar data than focal data that were taken further apart. 3) What were your random effects? 4) If the interaction with the age-squared term is not significant, do you need it in the model full model or would it be better to omit it? 5) Would it make sense to extract the P-value for the main effect of offspring sex, if interactions including offspring sex are not significant? You find no sex differences in social development of male and female offspring but maybe there is just not an interaction with age and/or maternal investment? Did you check main effects – you say you do, but it is unclear to me from Table 3?

Results

L212: suggestion: “…that immatures were involved in grooming with other…”

Table 3)

General comments: 1) Figure titles are usually placed below the figure – is this different in Animals? 2) Fig.1&2: Since you measured time spent in these behaviors, wouldn’t it be less confusing to talk about proportion of time spent grooming and playing on y axis? Similarly, the unit of maternal investment 0-1 needs to be explained in the figure description. 3) In your aim/hypothesis/prediction 1, you emphasize that you will focus on the first year of life (early differences). This aspect seems to be ignored in the analysis and results presentation.

Figure 1) Looking at the data maybe maternal investment even follows a quadratic function for males (low grooming with others when maternal investment is very low and very high)?

Discussion

L243-244: “Largely in line with our predictions (Prediction 2), we found differences between female and male immatures in the importance of maternal investment for their social integration in the group (Table 1).” In other words, you found that increased time mothers invested positively affected sons’ time spent playing and grooming with other group members, while daughters can reach comparable levels of playing and grooming with others independent of how much the mother invests in her daughter, right? So, males are more costly and needy? Why? Could it be that there have been different selective pressures on female and male offspring’ traits which are reflected in behavioral patterns/social development of both sexes early in live, e.g. maybe females become more independent from mothers and explore their close (social) environment earlier than males. But even if that is the case, you need to make a clearer case how more grooming, more co-feeding, more playing and contact involving mother and son can translate in spending more time in these behaviors with other group members in sons? I struggle to understand well the mechanism how one encourages/leads to the other. As mentioned before, it even seems a bit counterintuitive because theoretically, the more time the son spends with mum, the less time he has available for interacting with other group members. I am not doubting the data presented here but I am not convinced that your maternal investment measures are capturing the actual factors that drive the level of offspring social integration in the group even if you found some relationships and you did not necessarily aim to look for the drivers but simply for a link between both. I think other mechanism such as the social environment of the mothers, which determines exposure of offspring to other monkeys at young age, especially when youngsters’ travel still depends on mum – this could foster the establishment of bonds with other group members. Maybe mothers invest in males more in general (see later discussion) and those who invest most are high ranking and surrounded by more group members (see later discussion). Therefore, mothers who invest more in sons may also expose them to a richer social environment that fosters social integration. Speaking of social integration as part of your first discussion statement! Please let me also repeat here that the measures you used to define social integration are limited and lack some key components, such as how many social partners were involved and who were these social partners. The latter would also aid to discuss your finding addressed in lines L248-251 (“Similarly, we found no support to our prediction (Prediction 1) that immature males would be overall more likely than immature females to interact with other group members, and that these differences would increase when approaching sexual maturity) more meaningful in the light of a male-philopatric species, because the actual ID of interaction partners (related or not/ males or females) may matter and differ between female and male offspring. Maybe gaining skills in socializing with others is important for both sexes but for different reasons and with different social partners – e.g. if females want to integrate successfully into a new group, shouldn’t they also gain a set of social skills?

L248-251: what about the first year focus which was lined out in the first aim

L252-261: In my opinion, this paragraph also needs to be discussed much more carefully for the reasons mentioned above (are your maternal investment measures really responsible for social integration and the limitations of social integration indicators used in the study). I would turn this paragraph into a suggestion for future studies.

L262-269: Here you should discuss in more depth findings addressing your Prediction 1. Why did you not find any sex differences in time socializing with others in this male-philopatric species. See comment above!

L271-272: Speaking of social networks! This would have been indeed a great measure for social integration – how strong are male and female offspring social networks? If you have the data, I think such additional analysis would strengthen your study!

L272-286: You could avoid arguing here if you have data that exactly investigate this issue: social environment of mothers and how this affects social integration of their offspring. As said already, it is a weakness of your study that these two measures are lacking: 1) mother’s social environment and 2) key measures of social integration. Apologies for repeating myself, but if you have these data, please include them and run additional analysis. It would massively strengthen your study and add crucial missing pieces to make the “story” round.

L278-298: This picks up again the point addressed in L252-261. Maybe combine these paragraphs or move any evidence of sex-biased maternal investment to the introduction supporting the idea of your study why it is worth investigating if this sex-bias could translate into differences in social integration.

L299-308: Here again, having data on the sex of social partners would really help to better understand what is going on. Even if both sexes seem to integrate well, they may establish different social networks.

L329: I think your data do not support gradual because you did not show change over time male and female offspring throughout the first 5 years.

L333: you need to be more careful: with possible fitness return.

Reviewer 2 Report

The study found that maternal investment influences on offspring's social affiliation depending on their sex. As a subject it is interesting, however, the data collection and analyses are concerning. Also the definition of terms should be clearly mentioned throughout the manuscript, some of them only appear in the discussion part, not matching with the data analysis or results. Please see the comments below.  

Simple summary

L17: “more grooming and playing from non-mother group members for sons”

L18: the term “social development” is confusing because L15 says “no sex differences in the social development” please specify throughout the manuscript.

L19: remove “specifically”

Abstract

L30: How is maternal investment defined?

Introduction

L80: “for instance, spending more time in social interactions with them”? or what is the example of mother investment here?

L81: “and spend more time in social interactions with them”: Does it mean that mothers socially interact with their offspring more? Or mothers socially interacting with other social partners with the presence of their offspring around so that it eventually affects the offspring’s access to social partners?

Data analysis

L161-164: If proxies of maternal investment were collected from Feb 2022 only, what have you matched to the dependent variables between August 2021 to January 2022? Or have you generalized the maternal investment from Feb 2022 to June 2022 and gave that one value for each individual? Either case would be problematic since behaviours you are using as dependent variables were collected before the period you have collected the independent variable.

L179: For the model 3 (co-feeding), have you considered controlling for some ecological factors such as food availability or seasonality that might affect the co-feeding behaviour itself?

Discussion

L247-248: Where does this “probability of approaching” come from? Do you mean body touch from model 1? How do you use “(L159) time spent in body contact” and “probability of approaching” interchangeably? It is totally a different thing. There is nothing about “approaching” in M&Ms and results except in the abstract and discussion. Please clarify this.

L289: “male offspring”? or “sons”?

Conclusion

L333: How do you prove “with significant fitness returns”?

Reviewer 3 Report

This is a very interesting report

It does however raise an important question: who initiates the “parental care”, the mother or the offspring? In the data collection section you report that you continuously sampled, so you should have information as to who initiated body contact, nursing, grooming (which would be useful if grooming between partners is reciprocated in spider monkeys as reported in ref 67), touching and playing.

I bring this up as in spider monkeys where males are philopatric, male offspring who solicit/obtain more maternal care would be expected to have greater reproductive success (RS)  (Trivers, R. L. 1974 Parent-offspring conflict. Amer. Zool. 14, 249–264) to the extent that social integration is associated with greater RS. Can you address this in the current manuscript or are you planning to include such information in the paper in preparation that is cited as reference 64?

Discussion lines 262-270 is good

Other more technical comments:

You report observations between August 2021 and June 2022 but you only collected data on time in different forms of parental care from Feb to June of 2022. Was only the latter data included in your analyses? Do the # hours observed accurately reflect the number of focal samples used in the binomial analyses presented?

Looks like you were using alpha less than or equal to .05 to assess significance, but would be good to state this.

Line 296 replace “included” with “observed”

Line 324 would be good to include/name specific measures of physiological costs for different components of maternal investment that could/would be included in future studies

Round 2

Reviewer 2 Report

Thank you for revising your manuscript. Unfortunately, the concern regarding data collection and analysis raised by the other reviewer and me remained unsolved and poorly justified. Maternal investment can highly vary depending on the offspring's developmental stages. If the authors could not collect data on maternal investment at the beginning of the study, the dependent variables should at least match the same period (use data from Feb 2020). Current data analyses are problematic, and the results are unreliable, making it unacceptable to publish as it is. 

Author Response

For explorative purposes, we have re-run the analyses only including a subset of the data, as the Reviewer had originally proposed. The results largely confirm the results obtained with the whole data set, although confidence intervals are unsurprisingly much larger, and some models have some convergence issues, due to the smaller size of the dataset. Therefore, and after improving the models in line with the Editor’s suggestion, we are confident that our study provides a good despite preliminary description of the developmental patterns of social interactions in wild spider monkeys.